# Combined glycerol and sodium bicarbonate elicits improvements in fluid retention and blood buffering capacity

Ben Patrick◉*, Charles Urwin, Andrew C. Betik, William T. Jardine, Rhiannon Snipe, Monica Kelly, D. Lee Hamilton, Amelia Carr

School of Exercise and Nutrition Sciences, Deakin University, Burwood, Victoria, Australia

* b.patrick@deakin.edu.au

## Abstract

### Introduction

**Background:** It is unclear whether hyperhydration induced via glycerol, sodium bicarbonate or a combination of the two is most effective.

**Aim:** This study evaluated the effects of glycerol-, sodium bicarbonate-, and glycerol + sodium bicarbonate-induced hyperhydration on measures of fluid retention, blood buffering capacity, and gastrointestinal symptoms at rest, in comparison to a fluid-only control.

### Methods

Eleven healthy participants (six male) completed four trials (randomised, crossover design). Treatments consisted of glycerol-(G) = 1.0 g·kg⁻¹ body mass glycerol; sodium bicarbonate-(SB) = 0.3 g·kg⁻¹ body mass sodium bicarbonate; and combined glycerol and sodium bicarbonate supplementation-(G + SB) = 1.0 g·kg⁻¹ body mass glycerol + 0.3 g·kg⁻¹ body mass sodium bicarbonate; and a fluid only control-(CON) = 25 mL·kg⁻¹ body mass artificially sweetened water. At baseline and at 20 min intervals for 180 min, urine samples were collected and analysed for volume, colour, and specific gravity, and gastrointestinal symptoms and body mass were quantified. At 60 min intervals, capillary blood was analysed for pH, blood bicarbonate concentration and plasma volume change.

### Results

Compared to CON, fluid retention was significantly higher with G + SB from 120–180 min, SB from 120–140 min, and G from 160–180 min (p < 0.05), and accumulated urine volume was significantly lower with G + SB from 100–180 min, SB from 120–180 min, and G from 140–180 min (p < 0.05). There were no significant differences in fluid retention between G + SB, SB or G at any time point (p > 0.05). Compared to G

**Data availability statement:** The data from this project is available through Deakin University's research repository at the following DOI: https://doi.org/10.26187/deakin.29434217.v1.

**Funding:** Funding for this investigation was provided by the School of Exercise and Nutrition Sciences at Deakin University.

**Competing interests:** The authors have declared that no competing interests exist.

and CON, blood pH and blood bicarbonate concentration was significantly higher with G+SB and SB from 60–180 min ($p < 0.05$).

## Conclusions

Fluid retention improved in G, SB and G+SB compared to CON, although there were no additive effects with G+SB compared to SB or G. Buffering capacity was significantly higher with SB and G+SB compared to G and CON. Minimal GI symptoms were induced by G, SB, G+SB and CON.

## Introduction

Climate change continues to increase the temperatures in which international sporting events are held [1], therefore, strategies to protect athletes' health and performance capacity in the heat are of high importance. Exposure to challenging environmental conditions (ambient temperatures > 26°C and relative humidities > ~40% [2]) during exercise can increase sweat rates, making it difficult to match fluid losses with fluid ingestion, subsequently increasing the risk of dehydration (typically defined as loss of total body water below normal levels) and heat stress [3]. Compared to exercising in cooler conditions (e.g., ~10.5°C; 70% relative humidity), exercising in conditions with high environmental heat stress accelerates total body water losses and impedes endurance performance due to increased cardiovascular strain and impaired thermoregulation [3,4]. Compared to a euhydrated state (typically defined as normal total body water), dehydration can impair performance by 3–23%, with decrements increasing as environmental heat stress rises [5].

Water-induced hyperhydration (typically defined as an increase in total body water above normal levels) is ineffective at sustaining a hyperhydrated state, due to hypo-osmolality rapidly suppressing anti-diuretic hormone, resulting in increased urinary volume and fluid losses [6,7]. Supplementation with osmotic agents (e.g., glycerol and sodium ($Na^+$)), however, has been demonstrated as an effective strategy to improve hydration status and induce hyperhydration [8–11]. It has been established that hyperhydration supplements are typically emptied rapidly by the stomach to be absorbed and integrated into the fluid pools of the body, and may reduce urine production, increase fluid retention (defined as fluid intake minus urine volume ($U_{vol}$)), and increase plasma volume (PV) [9,11].

Hyperhydration interventions include glycerol (~1.0–1.2 grams per kilogram ($g \cdot kg^{-1}$) body mass (BM)) and sodium ($Na^+$) (~128–164 millimoles per liter ($mmol \cdot L^{-1}$)) [6,8–13], which contain osmotically active solutes that are readily absorbed into fluid compartments and have a relatively high osmotic pressure [8,14]. For glycerol-induced hyperhydration to increase and sustain total body water above euhydration levels, a glycerol dose higher than that present in food sources is required [11,15]. Glycerol is rapidly absorbed via passive diffusion in the gastrointestinal (GI) tract before being distributed amongst both intracellular and extracellular fluid compartments [8]. The compound containing $Na^+$ most widely investigated for its buffering

capacity is sodium bicarbonate ($NaHCO_3$) [15–19]. Sodium chloride (NaCl) has been established as an intervention to induce hyperhydration, as $Na^+$ can maintain or increase the secretion of anti-diuretic hormone, preventing the rapid increase in urine output that occurs with water induced hyperhydration [8–10,20].

Glycerol and $Na^+$ induce fluid retention via different mechanisms with glycerol retaining approximately 39% and NaCl 61% of ingested fluid 180 min post-ingestion [11]. However, when ingested in combination, their effects may be additive, in that 77% fluid retention at 180 min post ingestion has been observed [11]. Currently the only combined $Na^+$ and glycerol hyperhydration strategies to be fully investigated have utilised NaCl [11]. However, $NaHCO_3$ has recently been demonstrated as a source of $Na^+$ to induce hyperhydration [15]. Consequently, combining $NaHCO_3$ and glycerol may induce additive fluid retention, similar to that previously reported in combined NaCl and glycerol hyperhydration strategies, while also concurrently improving buffering capacity. Additionally, following isolated ingestion of both glycerol and $NaHCO_3$, GI symptoms have been reported [10,21,22]. Because GI symptom incidence and severity may be additive when glycerol and $NaHCO_3$ are co-ingested, and as GI discomfort could blunt the ergogenic benefits of increased hydration and/or buffering capacity [9,23], it is essential to quantify these symptoms when investigating co-ingestion. Therefore, the aim of this research is to establish a time course for measures of hydration (change in plasma volume (ΔPV), urine specific gravity (USG), urine volume ($U_{vol}$), urine colour ($U_{col}$), fluid retention, and BM)), buffering capacity (blood pH and blood ([$HCO_3^-$]), and GI symptoms (rating) in response to glycerol-(1.0 g·kg$^{-1}$ BM); G), $NaHCO_3$-(0.3 g·kg$^{-1}$ BM; SB), and combined glycerol (1.0 g·kg$^{-1}$ BM) and $NaHCO_3$-(0.3 g·kg$^{-1}$ BM) ingestion (G + SB), compared to a matched fluid dose (25 millilitres per kilogram (mL·kg$^{-1}$ BM).

## Methods

### Participants and recruitment

Eleven healthy adults (six males and five females; 27.1 ± 8.1 years, 177.4 ± 9.5 cm, 74.4 ± 12.6 kg), who were physically active (tier one (recreationally active: n = 6; tier two (trained): n = 4; tier three (highly trained): n = 1)) [24], with no current or past incidence of cardiovascular disease [11,15], completed the study. A twelfth participant commenced the study, however, withdrew for personal reasons. The inclusion criteria for participation in the study comprised healthy BM (body mass index within 18.5–24.9) [15], and aged between 18−50 years [6]. The exclusion criteria comprised using any medication for pre-existing conditions, abnormal renal function [15], prescribed diuretic or creatine use in the past 8 weeks [11,15], known allergies to either glycerol or $NaHCO_3$ [25], and a history of migraines [8], chronic headaches [8] or liver disease [26]. Participants were excluded if they answered 'Yes' to any of the first 6 questions of stage 1 of the Exercise and Sport Science Australia (ESSA) pre-screening tool or indicated to have >2 risk factors after completing stage 2 [27]. A sample size estimate was performed using STATA Version 18.0 (College Station, Texas, USA) (n = 10; power of 0.8, and alpha of .05). The estimate was performed using fluid retention data (glycerol + NaCl) obtained from Goulet et al. (mean difference of 200 mL; SD 200 mL) [11]. All participants were informed verbally and in writing of the nature of the investigation, including potential risks, and provided written consent prior to the collection of any experimental data. Recruitment commenced on the 15$^{th}$ of July 2024 and concluded on the 1$^{st}$ of December 2024. Ethical approval was provided by Deakin University Human Research Ethics Committee (2024−053).

### Overview

This study used a randomised, semi-counterbalanced, non-blinded, crossover design. Participants visited the laboratory five times for baseline testing and four experimental treatments: 1) glycerol (1.0 g·kg$^{-1}$ BM, G); 2) $NaHCO_3$ (0.3 g·kg$^{-1}$ BM, SB), 3) glycerol + $NaHCO_3$ (G + SB) and 4) equivalent fluid dose (25 mL·kg$^{-1}$ BM) with no active ingredient (CON). Measures of fluid retention, blood buffering capacity, and GI symptoms were collected (Fig 1). A wash-out period with a minimum of seven days was enforced between experimental treatments to eliminate any residual effects of $NaHCO_3$ supplementation on buffering capacity [15].

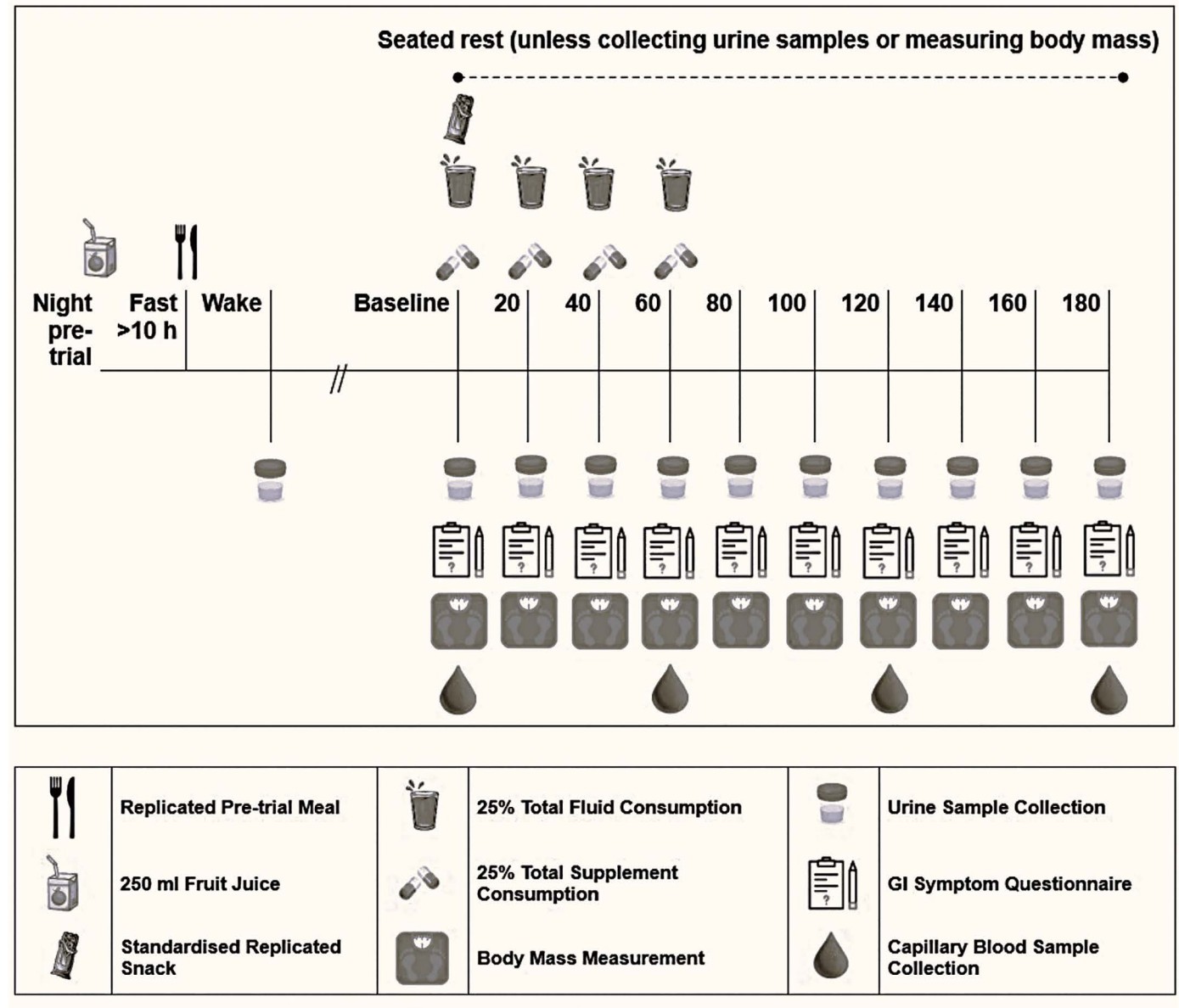

**Fig 1. Schematic overview of experimental trials.**

### Experimental testing and procedures

Prior to the collection of any data, an online meeting was held with potential participants to identify any exclusion criteria [25]. Participants arrived at the laboratory for all visits following a 10 h (overnight) fast. Height (cm) was measured using a stadiometer (Harpenden; Holtain Ltd; Crymych, UK) and BM (kg) was measured using scales (IC-UC-321; A&D Co., Ltd; Tokyo, Japan) with participants dressed in light, minimal clothing [15]. Participants were familiarised to the use of Easy Diet Diary (Version 6.0.28; Xyris Software; Brisbane, Australia).

Dietary intake for 24 h prior to each testing session was recorded using a smartphone application (Easy Diet Diary), with dietary and fluid intake replicated prior to each remaining experimental testing visit. To standardise hydration status,

participants were provided with a standardised drink (250 mL; Orange Fruit Drink; Golden Circle; Northgate, Queensland) the night before the trial. On the day prior to each experiment, caffeine, alcohol and light exercise were permitted until midday and the consumption of high $Na^+$ foods and the use of a spa or sauna was prohibited.

Participants arrived at a standardised time (the same time of day for each participant; between 7 and 10 AM) for each session to avoid diurnal variation in BM fluctuations [28]. On the morning of the trial, participants collected a midstream urine sample in a specimen jar (P5744SL; Techno Plas; Adelaide, Australia) upon waking to bring to the laboratory. The sample was assessed for USG via a refractometer (PAL-10S; ATAGO Co., Ltd; Tokyo, Japan) and $U_{col}$ was assessed by an eight-point scale [15,29,30]. Participants provided a full-void urine sample (724315; Greiner Bio-One; Singapore), which was weighed using scales (MII-2500; Universal Weight Electronics; Geldermalsen, The Netherlands) to the nearest gram, and assessed for baseline USG, $U_{vol}$ and $U_{col}$ [30]. A urine sample was requested from participants every 20 min following trial commencement (Fig 1). Participants attempted to provide a sample at each time point, but if unable to do so they notified the research team, and a value of 0 mL was recorded as the $U_{vol}$. Baseline BM was recorded.

In preparation for initial blood collection, participants were seated with their hand in a warm water bath for 7 min, allowing for postural stabilisation and arterialisation [30,31]. Two 100 microlitre (µL) capillary blood samples were collected and analysed for haemoglobin concentration (Hb), blood pH and [$HCO_3^-$] using a blood gas analyser (ABL800 Flex; Radiometer; Denmark) [32]. Two 75 µL capillary blood samples were collected and analysed for haematocrit via the microhaematocrit method ((Megafuge 8; Thermo Fisher Scientific; MA, USA) [33]). A validated questionnaire was used to measure GI symptoms [15,34,35].

The hyperhydration protocol consisted of a 180 min period sub-divided into nine 20 min segments (Fig 1). Participants remained seated unless providing urinary samples or BM measurements. In each of the first four 20 min segments, 25% of the treatment beverage volume was consumed within 5 min [15]. Participants consumed a standardised meal containing 1 g·kg$^{-1}$ BM of carbohydrate (Toast, High Fibre Wholemeal Bread; Coles; Melbourne, Australia) with jam (Strawberry Jam; Bega Group; Melbourne, Australia) at the beginning of the first 20 min segment within 10 min [15,34]. At the conclusion of each 20 min segment, the GI symptom questionnaire was completed [35], BM was measured, and $U_{vol}$ was collected, weighed, and analysed for USG and $U_{col}$ as previously described [15]. Capillary blood samples were taken every 60 min to monitor changes in blood pH, [$HCO_3^-$] Hb and Hct.

All treatments were administered using a sugar-free artificial sweetened beverage (Zero Sugar Cordial; Cottee's; Sydney, Australia), prepared according to the manufacturer's instructions, amounting to 25 mL·kg$^{-1}$ BM of fluid intake [15]. Each dose was prepared the day prior by a research team member, divided into four equal portions, and stored at room temperature [36]. Supplements had been batch tested for prohibited substances via a third-party auditing company [37]. Given that each supplement can independently increase the incidence and severity of GI symptoms, a lower glycerol dosage of 1.0 g·kg$^{-1}$ BM than most frequently reported in previous studies was utilised during combined supplementation [10,21,22]. A dose of 1.0 g·kg$^{-1}$ BM glycerol was used as it has been demonstrated to significantly increase fluid retention compared to a fluid bolus ingested alone [22]. The $NaHCO_3$ dose reported to increase buffering capacity and hydration status (0.3 g·kg$^{-1}$ BM) was used [15], combined with a fluid volume of 25 mL·kg$^{-1}$ BM [12,13,15].

Glycerol (Glycerol BP; Gold Cross; Melbourne, Australia) within the G and G+SB treatments was ingested as a 25 mL·kg$^{-1}$ BM solution for ingestion. For the SB and G+SB trials, the $NaHCO_3$ dose was contained within dissolvable gelatine tablets (Sodibic; Chemist Warehouse; Melbourne, Australia) and divided into four equal portions to be ingested with the fluid [15]. Participant BM was used to calculate the appropriate number of capsules to achieve a $NaHCO_3$ dosage of 0.3 g·kg$^{-1}$ BM.

Changes in PV were calculated using Hb and Hct values obtained via blood gas analysis:

$$\Delta PV,\ \%\ =\ 100\,((Hb0/\,Hbt)\ *\ ((1-Hctt)/(1-Hct0)))\ 1-100$$

Whereby subscripts 0 and t refer to measurements at baseline (0 min) and at time (60, 120, or 180 min post-ingestion) respectively [38].

Fluid retention (mL) throughout the trials were calculated as fluid ingestion minus $U_{vol}$ [39].

## Menstrual cycle control

Due to the impact of menstrual cycle phase on GI symptoms [40], and female sex hormones, which can impact body fluid distribution [41], menstrual cycle phase and the use of hormonal contraceptives was recorded. Of the five female participants, three were naturally menstruating and two were combined hormonal contraceptive pill users. Naturally menstruating female participants undertook all experimental sessions outside of days 1–3 of the menstrual cycle to minimise GI symptoms which can be reported during menses influencing the GI questionnaire [42–44]. Urinary ovulation tests (Ovulation Strip Tests + Strip Test Collection Cups; Fertility2Family; Glenorchy, TAS, Australia) were provided for participants and were used to identify the occurrence of ovulation for naturally menstruating females during each experimental trial. Naturally menstruating participants commenced ovulation testing on the day corresponding to their cycle length (days ~8–10 of the menstrual cycle) and continued testing daily for a maximum of 10 days or until a positive result was obtained [45]. Hormonal contraceptive users were provided with a hormonal contraceptive diary to track active hormone pill and inactive (sugar) pill usage, and withdrawal bleeding/ spotting. Oral contraceptive users underwent sessions during their active hormone phase [46,47].

## Statistical analyses

Dietary intakes were analysed using a nutritional intake analysis software (Foodworks Version 10; Xyris Pty Ltd; Brisbane; Australia) and expressed as absolute (g) and relative values (g·kg$^{-1}$ BM). Macronutrient and energy (kilojoules (kJ)) intake were also compared between each of the four treatments. Data normality was analysed by visually examining a quantile-quantile plot. To compare dietary intake across varied treatments, a one-way analysis of variance (ANOVA) was conducted using Jamovi software (Version 2.5; Jamovi; Sydney, Australia). A two-way repeated measures ANOVA (treatment x time) was conducted using Jamovi to compare outcome measures between treatments for measures of hydration (ΔPV, USG, accumulated fluid volume, $U_{col}$, BM, fluid retention), and buffering capacity (blood pH and [$HCO_3^-$]). Results were reported as interaction effects and main effects for treatment and time as we have done previously [15]. When significant treatment, time, or interaction effects were detected, a Bonferroni adjustment was manually applied to the significance threshold. Specifically, the accepted alpha level for all measures of 0.050 was divided by the number of comparisons performed. For measures every 60 min (blood pH, [$HCO_3^-$], ΔPV), 24 pairwise comparisons were performed, yielding an adjusted alpha level of 0.0021 (0.050/24). For measures every 20 min (USG, accumulated fluid volume, $U_{col}$, BM, fluid retention), 60 pairwise comparisons were performed, yielding an adjusted p value of 0.0008 (0.050/60). For each GI symptom, the total point scores for severity were summed across all data collection time points and analysed using a Friedman's two-way ANOVA by ranks test with a Bonferroni post hoc pairwise comparison in SPSS Statistics (Version 29; IBM Corporation, NY, USA) [48].

# Results

## Participant characteristics and dietary standardisation

There were no significant differences between treatments in dietary Na$^+$ (g, g·kg$^{-1}$ BM), energy intake (kJ), macronutrient intake (carbohydrate, protein and fat; g, g·kg$^{-1}$ BM) or total water intake (mL, mL·kg$^{-1}$ BM) the day prior to experimental trials (p > 0.050) (Table 1).

## Hydration status

There were no significant differences in on-waking $U_{col}$ or USG between each treatment (p > 0.050; Table 2). There were no significant differences in BM, $U_{vol}$, USG, or $U_{col}$ between each experimental treatment on-arrival to the laboratory

**Table 1. Mean±SD dietary intake for participants (n = 11) the day prior to each trial.**

| Nutritional Intake | | CON | G | SB | G+SB | p-value |
|---|---|---|---|---|---|---|
| Na⁺ | g | 3.4±2.5 | 3.4±2.5 | 3.6±2.4 | 3.3±2.5 | 0.997 |
| | g·kg⁻¹ BM | 0.05±0.03 | 0.05±0.03 | 0.05±0.03 | 0.04±0.03 | |
| Energy (kJ) | | 12323.5±4805.0 | 12553.3±4582.1 | 12389.3±4699.1 | 12403.2±4714.2 | 0.999 |
| Carbohydrate | g | 353.4±97.4 | 345.6±92.7 | 352.9±96.3 | 357.2±89.3 | 0.997 |
| | g·kg⁻¹ BM | 4.8±1.3 | 4.6±1.2 | 4.7±1.3 | 4.8±1.2 | |
| Protein | g | 138.5±81.2 | 153.2±87.1 | 137.7±79.2 | 138.6±80.3 | 0.985 |
| | g·kg⁻¹ BM | 1.9±1.1 | 2.1±1.2 | 1.9±1.1 | 1.9±1.1 | |
| Total Fat | g | 102.8±57.1 | 105.0±55.6 | 105.0±54.7 | 103.4±54.2 | 0.999 |
| | g·kg⁻¹ BM | 1.4±0.8 | 1.4±0.7 | 1.4±0.7 | 1.4±0.7 | |
| Total Water | mL | 3556.1±859.8 | 3266.4±777.0 | 3415.2±913.6 | 3581.2±763.8 | 0.883 |
| | mL·kg⁻¹ BM | 47.8±11.6 | 42.9±10.4 | 45.9±12.3 | 48.1±10.3 | |

**Table 2. Mean±SD participant body mass and urinary measures of hydration status at baseline for each trial (n = 11).**

| Variables | | CON | G | SB | G+SB | *p* value |
|---|---|---|---|---|---|---|
| BM (kg) | | 71.9±12.3 | 72.2±12.4 | 71.9±12.2 | 71.4±12.1 | 0.998 |
| $U_{vol}$ (mL) | | 81.2±79.1 | 67.5±54.6 | 59.5±37.6 | 70.4±57.2 | 0.858 |
| USG (au) | Upon waking | 1.020±0.005 | 1.019±0.007 | 1.019±0.006 | 1.019±0.008 | 0.983 |
| | Laboratory arrival | 1.017±0.006 | 1.020±0.005 | 1.020±0.007 | 1.020±0.004 | 0.590 |
| $U_{col}$ (au) | Upon waking | 5.5±1.0 | 5.3±0.9 | 5.4±1.0 | 5.5±0.9 | 0.918 |
| | Laboratory arrival | 5.2±1.0 | 5.4±0.7 | 5.3±0.9 | 5.8±0.6 | 0.284 |

($p > 0.050$; Table 2.0). Participants were considered euhydrated at the beginning of these trials (USG within 1.013–1.024 au) [49]. Based on $U_{col}$ data however, participants were considered hypohydrated (within 4–6 au) at trial commencement [50].

For BM, there were no significant interaction (treatment x time; $p = 1.000$) or main effects for treatment ($p = 0.992$) or time ($p = 1.000$) (Fig 2A). There was a significant interaction effect (treatment x time; $p < 0.001$), and significant main effects for treatment ($p < 0.001$) and time ($p < 0.001$) for accumulated $U_{vol}$ (Fig 2B). There were no significant differences between treatments during the first 80 min post-ingestion ($p > 0.0008$), and no significant differences between G+SB, SB and G during the entire 180 min post-ingestion trial period ($p > 0.0008$). Compared to CON, accumulated $U_{vol}$ was significantly higher with G+SB from 100–180 min post-ingestion ($p < 0.0008$), SB from 120–180 min post-ingestion ($p < 0.0008$), and G from 140–180 min post-ingestion ($p < 0.0001$). For fluid retention, there was a significant interaction effect (treatment x time; $p < 0.001$), and significant main effects for treatment ($p < 0.001$) and time ($p < 0.001$) (Fig 2C). There were no significant differences between treatments during the first 100 min post-ingestion ($p > 0.0008$), and no significant differences between G+SB, SB and G during the full 180 min post-ingestion trial period ($p > 0.0008$). Compared to CON, fluid retention was significantly higher with G+SB from 120–180 min post-ingestion ($p < 0.0008$), SB from 140–180 min post-ingestion ($p < 0.0008$), and G from 160–180 min post-ingestion ($p < 0.0001$).

There was a significant interaction effect (treatment x time; $p < 0.001$), and significant main effects for treatment ($p < 0.001$) and time ($p < 0.001$) for USG (Fig 3A). There were no significant differences between treatments during the first 100 min post-ingestion ($p > 0.0008$), and no significant differences between G+SB, SB and G during the entire 180 min post-ingestion trial period ($p > 0.0008$). Compared to CON, USG was significantly higher with G+SB and G from 120–180 min post-ingestion ($p < 0.0008$), and SB from 140–180 min post-ingestion ($p < 0.0001$). For $U_{col}$, there was no significant

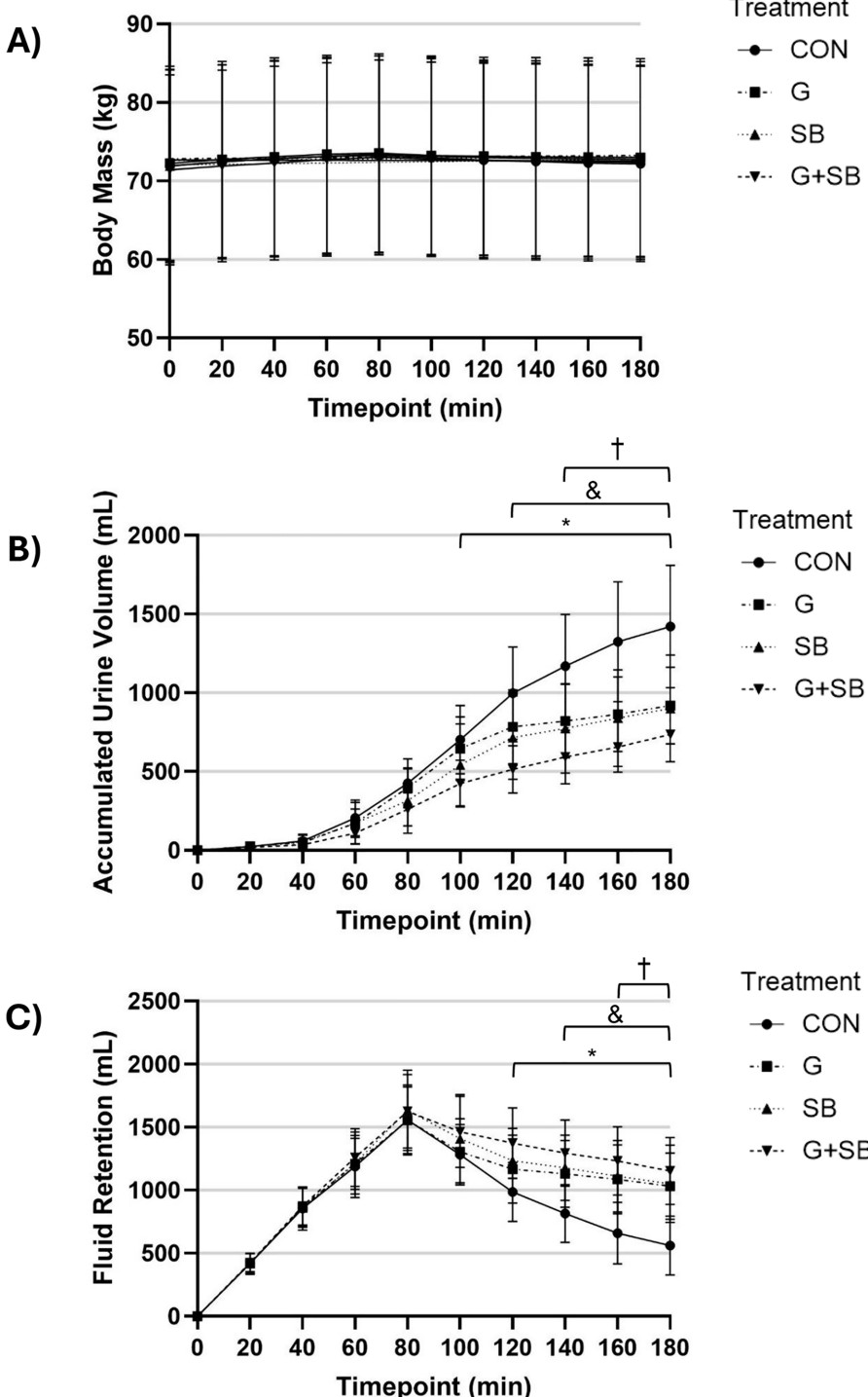

**Fig 2. Mean and 95% confidence interval (CI) for A) Body mass, B) Accumulated urine volume, and C) Fluid retention at each time point following either a glycerol (G), sodium bicarbonate (SB), combined glycerol and sodium bicarbonate (G+SB) or fluid only (CON) treatment (n = 11).** * $p < 0.0008$, significant difference between G+SB and CON. $^{\&}$ $p < 0.0008$, significant difference between SB and CON. $^{\dagger}$ $p < 0.0008$, significant difference between G and CON.

A)

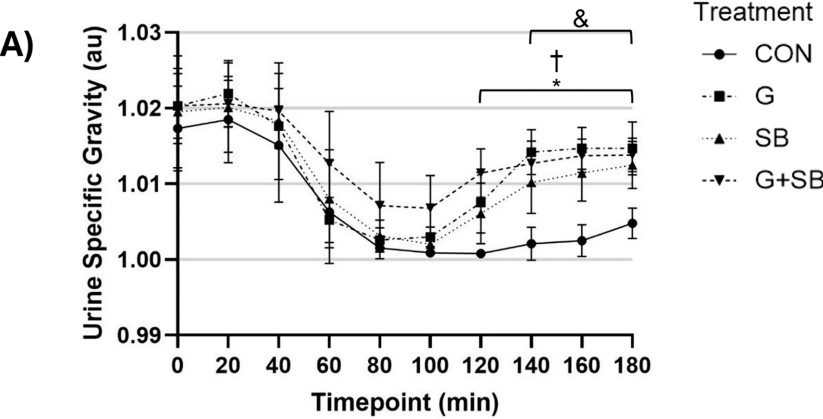

B)

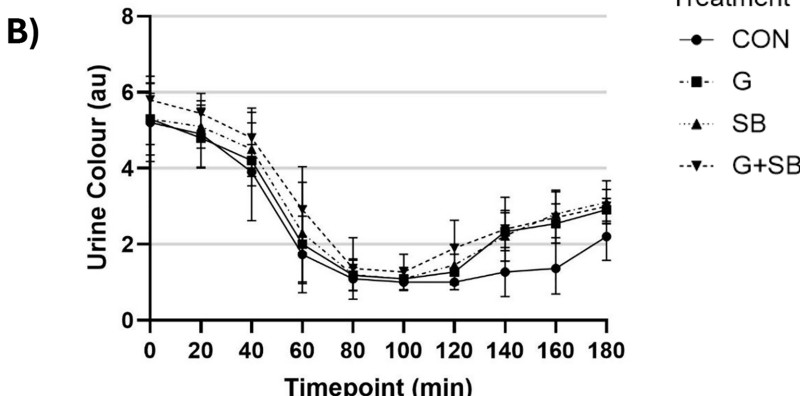

C)

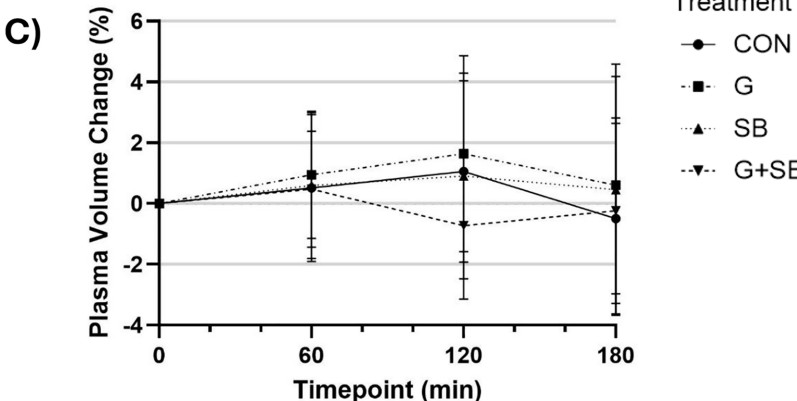

**Fig 3. Mean and 95% CI for A) Urine specific gravity, B) Urine colour, and C) Plasma volume change at each time point following either a glycerol (G), sodium bicarbonate (SB), combined glycerol and sodium bicarbonate (G+SB) or fluid only (CON) treatment (n = 11).** * $p < 0.0008$, significant difference between G+SB and CON. & $p < 0.0008$, significant difference between SB and CON. † $p < 0.0008$, significant difference between G and CON.

interaction effect (treatment x time; p=0.211) (Fig 3B). There was no significant interaction (treatment x time; p=0.922), or main effects for treatment (p=0.392) or time (p=0.437) for ΔPV (Fig 3C).

## Blood buffering capacity

There was a significant interaction effect (treatment x time; p<0.001), and significant main effects for treatment (p<0.001) and time (p<0.001) for blood pH and blood [$HCO_3^-$] (Fig 4). Blood pH and blood [$HCO_3^-$] was significantly higher with G+SB and SB compared to CON from 60-, 120- and 180-min (p<0.0021) post-ingestion. Blood pH and blood [$HCO_3^-$] was significantly higher with G+SB and SB compared to G at 60-, 120- and 180-min post-ingestion (p<0.0001).

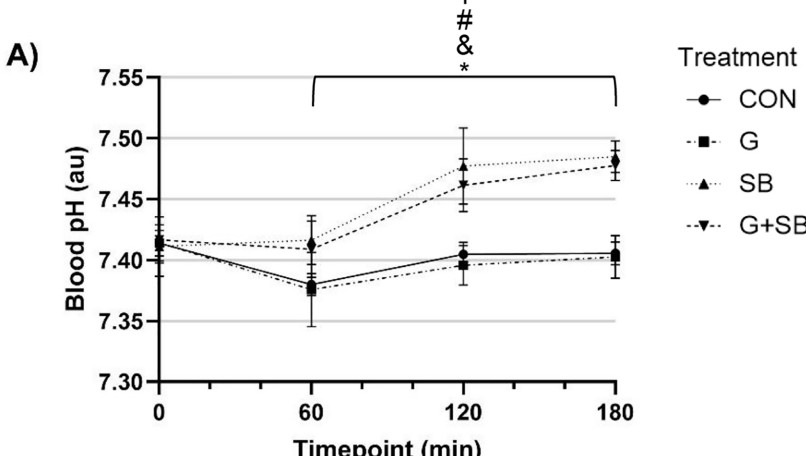

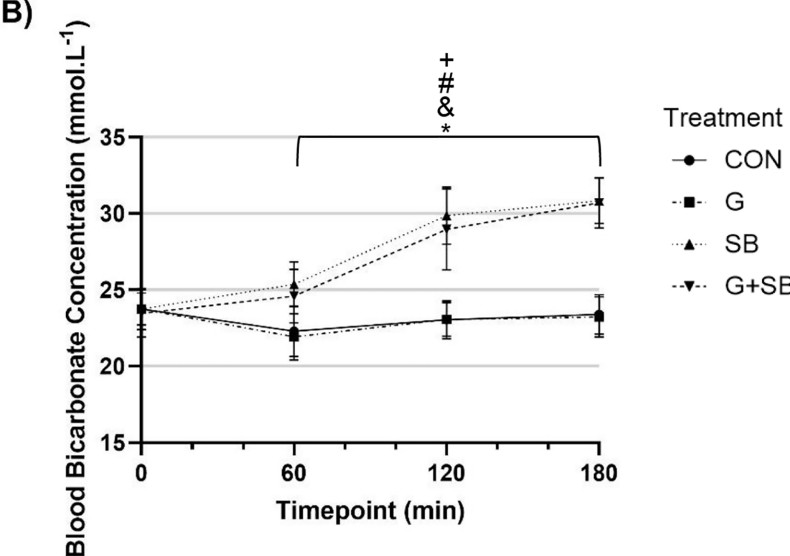

**Fig 4. Mean and 95% CI for A) blood pH, and B) blood bicarbonate concentration at each time point following either a glycerol (G), sodium bicarbonate (SB), combined glycerol and sodium bicarbonate (G+SB) or fluid only (CON) treatment (n=11).** & p<0.0021, significant difference between SB and CON. # p<0.0021, significant difference between G+SB and G. + p<0.0021, significant difference between SB and G.

## Gastrointestinal symptoms

There were no significant differences between treatments for individual, upper, lower, other and total GI symptom severity ($p > 0.050$; Table 3; S1 Table).

## Discussion

This study evaluated the effects of G, SB, and G+SB, and CON on hydration status, blood buffering capacity, and GI symptoms. This is the first study to our knowledge to investigate the combined hyperhydration potential of glycerol and $NaHCO_3$. The novelty of this study lies primarily in the key physiological insights gained into the timing responses of hydration status, blood buffering capacity, and GI symptoms following combined glycerol and $NaHCO_3$ hyperhydration. Future research will be able to utilise the timing-specific insights gathered from our study and integrate exercise testing. Despite no significant fluid retention differences between G, SB and G+SB, each experimental treatment induced significantly greater fluid retention compared to CON, and there were no significant differences in BM were reported across treatments at any time point. However, the G+SB treatment did elicit and sustain an improved fluid retention compared to CON at an earlier timepoint than SB and G. Additionally, G+SB improved blood buffering capacity (increased blood pH and blood $[HCO_3^-]$) although similar results were observed with SB. Importantly, the increased fluid retention and buffering

**Table 3. Incidence and severity of upper, lower, other and total gastrointestinal symptoms, following G, SB, G+SB and CON.**

| | CON | G | SB | G+SB | Friedman's p-value |
|---|---|---|---|---|---|
| **Mean total severity and range of total severity between individual participants[a]** | | | | | |
| Upper GI Symptoms | 21.7 (0-94) | 9.9 (0-83) | 5.0 (0-16) | 6.0 (0-40) | 0.797 |
| Lower GI Symptoms | 5.4 (0-22) | 5.2 (0-20) | 2.5 (0-12) | 2.7 (0-16) | 0.353 |
| Other GI Symptoms | 8.6 (0-71) | 8.4 (0-71) | 5.5 (0-27) | 12.8 (0-80) | 0.602 |
| Total GI Symptoms[d] | 35.7 (0-177) | 23.5 (0-104) | 13.0 (0-46) | 20.6 (0-98) | 0.435 |
| **Incidence (%) of GI symptoms (rating ≥1 at any time point)[b]** | | | | | |
| Upper GI Symptoms | 53.6 | 36.4 | 81.8 | 63.6 | NA |
| Lower GI Symptoms | 45.5 | 45.5 | 36.4 | 36.4 | NA |
| Other GI Symptoms | 36.4 | 36.4 | 36.4 | 36.4 | NA |
| Total GI Symptoms | 72.7 | 63.6 | 100 | 81.8 | NA |
| **Incidence (%) of severe GI symptoms (rating ≥5 at any time point)[c]** | | | | | |
| Upper GI Symptoms | 45.5 | 18.2 | 18.2 | 9.1 | NA |
| Lower GI Symptoms | 27.3 | 18.2 | 18.2 | 18.2 | NA |
| Other GI Symptoms | 18.2 | 9.1 | 18.2 | 27.3 | NA |
| Total GI Symptoms | 45.5 | 45.5 | 36.4 | 27.3 | NA |

[a]Total severity values are the sum of all reported GI symptoms at all measured time points from the relevant session. For total severity for upper, lower, and other GI symptoms, the maximum possible value was 700 (7 symptoms, reported 10 times, each with a maximum rating of 10), 900 (7 symptoms, reported 10 times, each with a maximum rating of 10), and 300 (7 symptoms, reported 10 times, each with a maximum rating of 10) respectively. For total severity for total GI symptoms, the maximum possible value was 1900 (19 symptoms, reported 10 times, each with a maximum rating of 10).

[b]Incidence (%) of GI symptoms indicates the percentage of participants which reported a rating ≥1 for the corresponding GI symptom and was calculated as: (number of participants who rated ≥1 for at least one symptom/sample size of 11) × 100).

[c]Incidence (%) of severe GI symptoms indicates the percentage of participants which reported a rating ≥5 for the corresponding GI symptom and was calculated as: (number of participants who rated ≥5 for at least one symptom/sample size of 11) × 100).

[d]Total GI symptoms summarise upper, lower and other GI symptoms. Upper GI symptoms: belching, heartburn, bloating, stomach pain, urge to regurgitate, regurgitation, and projectile vomiting, Lower GI symptoms: flatulence, lower abdominal bloating, urge to defecate, left and right intestinal pain, and defecation. Other GI symptoms: nausea and dizziness.

NA = not applicable. GI = Gastrointestinal.

capacity compared to CON were present without a concurrent increase in GI symptoms in the experimental conditions when compared with CON.

## Body mass, fluid retention and urine specific gravity

In the current study, there was no significant difference in BM across treatments at any time point. However, a significantly lower accumulated $U_{vol}$ was observed with G+SB compared to CON at 100−180 min post-ingestion, with SB compared to CON at 120−180 min post-ingestion, and with G compared to CON at 140−180 min post-ingestion, indicating increased fluid retention. Similar findings were reported by Sims et al., who used a lower fluid volume (10 mL·kg$^{-1}$ BM) but a similar $Na^+$ (NaCl and sodium citrate) concentration of 164 mmol·L$^{-1}$ (142.8 mmol·L$^{-1}$ used in the current study) [13]. Similarly, Siegler et al. used a similar $Na^+$ ($NaHCO_3$) concentration of 128 mmol·L$^{-1}$ with the same fluid volume and reported no significant differences in BM across treatments [15]. No significant BM differences across all treatments may indicate that BM measures lack the sensitivity to detect hyperhydration, and therefore other hydration measures (e.g., fluid retention) may be more informative in identifying these hydration changes.

In the current investigation, fluid retention peaked at 80 min post ingestion for all treatments. Compared to CON, fluid retention was significantly higher with G+SB from 120−180 min post-ingestion, with SB at 140−180 min post-ingestion, and with G at 160−180 min post-ingestion. At 180 min post-ingestion, fluid retention of the initial ingested fluid volume was 31.6% for CON, compared to 58.1% for G, 59.1% for SB, and 64.8% with G+SB. Goulet et al. reported 77% fluid retention at 180 min post-ingestion of ~24.9 mL·kg$^{-1}$ BM fluid and a combined treatment of glycerol (~1.16 g·kg$^{-1}$ BM) and of $Na^+$ (NaCl; 128 mmol·L$^{-1}$) [11]. In the same study, 61% fluid retention was reported at 180 min post-ingestion with $Na^+$ (NaCl; 128 mmol·L$^{-1}$) and 39% with glycerol (~1.16 g·kg$^{-1}$ BM) consumed in isolation [11]. Savoie et al. observed fluid retention of 59% with $Na^+$ (NaCl; 128 mmol·L$^{-1}$) and 41% with glycerol (~1.13 g·kg$^{-1}$ BM), with a fluid intake of ~24.3 mL·kg$^{-1}$ BM [6]. Previous studies reported that the lower $Na^+$ concentration derived from NaCl (compared with that derived from $NaHCO_3$ within the current study) was associated with an increase in BM [6], and a higher glycerol dosage was associated with lower fluid retention [11]. In the current study, USG increased with G, SB and G+SB compared to CON, which would be expected due to simultaneously decreased $U_{vol}$ and increased fluid retention increasing urine concentration [51]. Future research should therefore directly compare the effects of differing $Na^+$ sources on fluid retention. The combined ingestion of 0.3 g·kg$^{-1}$ BM $NaHCO_3$ and 1.0 g·kg$^{-1}$ BM glycerol elicited minimal GI symptoms in the current study, therefore a higher glycerol dosage (~1.2 g·kg$^{-1}$ BM) should also be investigated in future research. This specific combination with the glycerol provided at a higher dose may induce increased fluid retention in glycerol treatments compared with that observed in the current investigation, although the combination needs to be specifically tested as it may also elicit an increased incidence and severity of GI symptoms. Moreover, compared to G and SB, the current study suggests G+SB may induce more rapid fluid retention than when either supplement is ingested below, potentially making combined supplementation with glycerol and $NaHCO_3$ more effective for events where the time available prior to the start of the event for supplement ingestion is more limited (e.g., early morning starts for major city marathons).

## Plasma volume

There was no significant difference in ΔPV across treatments at any time point in the current study. Similarly, one previous study reported no difference in ΔPV following sodium induced hyperhydration, however, used slightly lower fluid volumes (16−17 mL·kg$^{-1}$ BM) and sodium concentrations (60 mmol·L$^{-1}$) than that of the current investigation [21]. Some previous studies have reported (as with the current investigation) changes in some but not all hydration measures [13,15]. One potential explanation for differences between the current investigation and Siegler et al., who reported a significant change in PV, but no significant change in BM, could be due to slight differences in Hb results produced by different analysers, despite both studies using valid instrumentation (ABL800 Flex, Radiometer and i-STAT 1 Wireless, Abbott, respectively) [15]. Additionally, in other previous studies where an increase in ΔPV was reported [6,11], a different analyser (HemoPoint

H2, EKF Diagnostics) than that used in the current study was used to generate Hb results. Therefore, future research needs to more thoroughly investigate and compare the results of different analysis methods that can be used to determine Hb.

## Blood pH and [HCO$_3^-$]

Blood pH and blood [HCO$_3^-$] was significantly higher with G + SB and SB compared to CON and G at all time points (60-, 120- and 180-min post-ingestion). For G + SB and SB, both blood pH and blood [HCO$_3^-$] peaked at 180 min post ingestion (pH of 7.48 ± 0.01 au and 7.49 ± 0.01 au respectively; [HCO$_3^-$] of 30.69 ± 1.65 and 30.82 ± 1.49 mmol·L$^{-1}$ respectively). This outcome is consistent with earlier research which reported similar increases and time points with 128 mmol·L$^{-1}$ of Na$^+$ from NaHCO$_3$ with 25 mL·kg$^{-1}$ BM fluid [15]. Previous work by Carr et al. [34], investigating the effects of 0.3 g·kg$^{-1}$ BM NaHCO$_3$, observed peak blood [HCO$_3^-$] values between 90−150 min post-ingestion. Compared to our investigation, Carr et al. had more frequent measures of blood [HCO$_3^-$] (every 30 min vs 60 min), allowing for a more precise blood [HCO$_3^-$] time course to be reported [34]. Measures of blood pH with G + SB and SB peaked in the current investigation at 0.07 and 0.06 au higher than baseline respectively 180 min post-ingestion. Also at 180 min post-ingestion, blood [HCO$_3^-$] with G + SB and SB peaked at 7.1 and 7.2 mmol·L$^{-1}$ higher than baseline respectively. Despite peaking at 180 min post-ingestion however, these increases exceeded the thresholds suggested to induce ergogenic benefits during high intensity and short duration exercise at 120 min post-ingestion (increase of blood [HCO$_3^-$] >6.0 mmol·L$^{-1}$ and blood pH >0.05 au) [23,34]. As peak blood pH and [HCO$_3^-$] values were observed at the final time point measured (180 min) in the current study, these markers of buffering capacity may continue to increase after the 180 min post-ingestion time point, as has been demonstrated with sodium citrate research [52]. Future research could therefore utilise longer trials with more frequent measures of blood pH and [HCO$_3^-$] to better identify time to peak, which could in turn more precisely inform the ingestion timing when implementing glycerol and NaHCO$_3$ hyperhydration strategies prior to competition.

## Gastrointestinal symptoms

Although some GI symptoms were observed within the current study (64–100% incidence rate), most symptoms were very minor in severity and frequency and there were no significant differences observed across treatments. The mean severity of total GI symptoms within all treatments was low (35.7, 23.5, 13.0 and 20.6 au for CON, G, SB and G + SB respectively), relative to the maximum possible value (1900 au). Goulet and colleagues similarly reported low abdominal discomfort severity at the end of the 180 min rest period following individual and combined glycerol and Na$^+$ (NaCl) ingestion [11]. The minimal GI symptoms reported in the current study may be at least partially explained by participants coingesting NaHCO$_3$ with a carbohydrate meal, a strategy reported to reduce the incidence of GI symptoms in comparison to when NaHCO$_3$ is ingested without a carbohydrate meal [34]. Early studies which reported GI symptoms typically used different types of ingestion protocols, such as those with NaHCO$_3$ prepared in a solution and without a coingested carbohydrate meal [53]. Over the past two decades however, as the practice of ingesting NaHCO$_3$ via capsules and with a carbohydrate meal has been more frequently utilised, NaHCO$_3$ studies have more commonly reported low incidence and severity GI symptoms [54]. Exercise alone can induce GI symptoms [55] which could interact with the minor symptoms observed in this study, therefore future research should therefore quantify the gastrointestinal symptoms observed after G + SB in the context of different types of exercise tests that simulate real-world competitive events

## Limitations and future research

In the current investigation, the hyperhydration protocols were not followed by an exercise trial, limiting our ability to assess the effects of G + SB, SB and G supplementation on performance. However, the current study was designed to isolate the ingestion protocol and responses to each condition without potentially confounding measurements of hydration status, buffering capacity or GI symptoms with the addition of exercise. Future studies may therefore include exercise

testing and quantify the effects of the types of hyperhydration strategies examined in the current study on exercise performance.

In the current study, the 'silver' tier methodology reported by Smith et al. [47] was implemented to document the female menstrual cycle phase and use of hormonal contraceptives. To examine the influence of the female menstrual cycle phase on hydration status and fluid retention [56,57], future investigations may use 'gold' standard methodologies, which incorporate the combination of calendar counting, ovulation testing, establishing a hormonal profile from blood sample analysis, and the collection of outcome measures across two cycles [47].

## Practical applications

Our results indicate that G + SB, SB and G are all useful for inducing fluid retention. Moreover, after G + SB ingestion, there was a sustained improvement in fluid retention compared to CON, which was also observed at an earlier timepoint than SB and G (120 min with G + SB compared to 140 min (SB) and 160 min (G)), with the added benefit of increased blood buffering capacity and without increasing GI symptoms. A potential practical application of our findings is in the time period immediately before competition, which is often short on the day of major events for endurance competitions. For instance, early morning starts are common for open water swimming, triathlon, marathon/ultra marathon and race-walking events, making G + SB a potentially more feasible hyperhydration strategy than SB or G for athletes competing in these events. The improvements in buffering capacity observed in this study suggest that G + SB and SB may be beneficial in some specific competitive events where adequate hydration status and buffering capacity may be beneficial (e.g., in the context of sprint finishes in road cycling events which also require prolonged effort [58]), or occupational contexts (e.g., shipboard firefighting performed by Navy personnel [59]).

## Conclusion

In the current investigation, glycerol-(1.0 g·kg$^{-1}$ BM; G), sodium bicarbonate-(0.3 g·kg$^{-1}$ BM; SB), and glycerol (1.0 g·kg$^{-1}$ BM) + sodium bicarbonate-(0.3 g·kg$^{-1}$ BM) (G + SB) improved fluid retention to a greater extent compared control (25 mL·kg$^{-1}$ BM; CON). Combined glycerol and sodium bicarbonate ingestion and isolated glycerol and sodium bicarbonate treatments improved fluid retention compared to control. Whilst the peak values observed do not indicate an additive benefit of sodium bicarbonate and glycerol ingestion when compared with each supplement in isolation, the time course observed suggests that combined glycerol and sodium bicarbonate may be beneficial when it is necessary to commence hyperhydration closer to the event start time.

## Supporting information

**S1 Table. The incidence and severity of all individual, upper, lower, other and total gastrointestinal symptoms in response to 180 min of rest following either G + SB, SB, G or CON.**
(DOCX)

## Author contributions

**Data curation:** Ben Patrick.

**Formal analysis:** Ben Patrick.

**Methodology:** Amelia Carr.

**Project administration:** Ben Patrick.

**Supervision:** Charles Urwin, William T Jardine, Rhiannon Snipe, Monica Kelly, D. Lee Hamilton, Amelia Carr.

**Writing – original draft:** Ben Patrick.

**Writing – review & editing:** Charles Urwin, Andrew C Betik, William T Jardine, Rhiannon Snipe, Monica Kelly, D. Lee Hamilton, Amelia Carr.

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
