## [Decision Letter · Decision Letter 0]

6 Oct 2025

Dear Dr. Patrick,

Thank you for submitting your manuscript to PLOS ONE. After careful consideration, we feel that it has merit but does not fully meet PLOS ONE’s publication criteria as it currently stands. Therefore, we invite you to submit a revised version of the manuscript that addresses the points raised during the review process.

We look forward to receiving your revised manuscript.

Kind regards,

William M. Adams

Academic Editor

PLOS ONE

Journal Requirements:

When submitting your revision, we need you to address these additional requirements

Reviewer's Responses to Questions

**Comments to the Author**

1. Is the manuscript technically sound, and do the data support the conclusions?

Reviewer #1: Yes

Reviewer #2: Partly

2. Has the statistical analysis been performed appropriately and rigorously?

Reviewer #1: Yes

Reviewer #2: Yes

3. Have the authors made all data underlying the findings in their manuscript fully available?

Reviewer #1: Yes

Reviewer #2: Yes

4. Is the manuscript presented in an intelligible fashion and written in standard English?

Reviewer #1: Yes

Reviewer #2: Yes

Reviewer #1: 1) Your study combines glycerol with sodium bicarbonate (NaHCO₃) for hyperhydration, which you note has not been directly investigated before. Could you more explicitly highlight what new insights this combination provides compared to the well-studied glycerol + NaCl strategy?

2) How does your finding that G+SB accelerates fluid retention (but does not enhance peak retention compared to SB or G alone) advance the field in practical terms? For example, could this be positioned as a novel solution for athletes with limited pre-competition preparation time?

3) Could you clarify whether the novelty of your study lies primarily in the physiological insight (timing and buffering interactions) or in the applied context (sport, occupational use), and how this differs from prior work?

4) The study reports no changes in plasma volume despite significant differences in fluid retention. Could you elaborate on why plasma volume may not have reflected fluid retention differences, and whether methodological sensitivity may be a factor?

5) You used a glycerol dose (1.0 g·kg⁻¹ BM) that is lower than some prior studies. Could you justify this choice more explicitly in terms of expected efficacy versus GI tolerance, and whether higher doses might have altered the outcomes?

6) Since no additive benefit was observed in peak fluid retention for G+SB compared to single agents, do you think the practical advantage of G+SB is mainly about timing of onset? If so, would you recommend it over SB alone in competitive practice?

7) You note that GI symptoms were minimal across conditions. Could you provide more context on whether this is consistent with prior SB-only studies, where GI issues are often reported? What might explain the reduced severity in your cohort?

Overall thoughts:

This manuscript addresses an important and timely question regarding hyperhydration strategies in sport and occupational contexts, specifically the combined ingestion of glycerol and sodium bicarbonate. To my knowledge, this is the first study to directly examine the effects of this combination on fluid retention and blood buffering capacity, and the findings offer valuable practical insights for athletes competing in hot environments and situations where pre-event preparation time is limited. The study is well-designed, with a randomized crossover approach, clear methodology, and careful monitoring of hydration, buffering markers, and gastrointestinal symptoms. The results are robust, showing that both glycerol and sodium bicarbonate enhance fluid retention compared to control, and that the combination accelerates retention and enhances buffering capacity without worsening GI symptoms. These findings make a meaningful contribution to the literature on hydration strategies and performance support. Overall, I find the study scientifically sound, ethically conducted, and appropriately reported. With minor adjustments to data availability and interpretation, this manuscript will make a valuable addition to the literature and I recommend it for publication.

Reviewer #2: In this study, investigators sought to assess the potential additive effects of sodium bicarbonate and glycerol on fluid retention and blood buffering parameters in physically active adults. Overall, fluid retention was increased with glycerol and sodium bicarbonate from 120-180 minutes post ingestion compared to control, promoting a slightly earlier time course for fluid retention compared to sodium bicarbonate and glycerol alone. This reviewer commends the authors for a well thought out study and offer the following recommendations to enhance the manuscript.

Introduction:

Line 72 add “and” before “increase plasma volume”.

Remove the double citation block on line 76. It is unclear if it should be (7, 17) or (11-16).

Remove double citation block in line 80.

You can separate the paragraphs for each hyperhydration agent.

The introduction could benefit from a brief discussion of the potential drawbacks of NaHCO3 on gastrointestinal distress, thereby supporting your decision to include this measurement.

Methods

Line 195 appears redundant to line 175-176

Line 205 – I recommend flipping the wording to make it clear that the subscript 0 refers to baseline, and subscript t refers to measurements at each subsequent timepoint, respectively.

Lines 210-226 – You captured important information regarding menstrual cycle. Although you excluded days 1-3 of the menstrual cycle to minimize the influence of hormonal fluctuations on GI symptoms, was the ovulation test merely used to determine participants were not anovulatory? I am a little confused by the decision to only exclude this time rather than isolate to a specific phase, particularly when some of the authors of the study previously found a more pronounced increase in body mass in phase 4 of the menstrual cycle: https://pubmed.ncbi.nlm.nih.gov/39591960/

Line 246 – 250 – Rewrite this sentence for clarity; the second half can be omitted.

Results

Line 290 - why are both p>0.050 and the exact the p-values reported?

The authors spend majority of the results discussing the difference between each beverage and the control condition. But the main effect of interest is any difference between G+SB and SB or G alone. It may be worth explicitly stating these null differences between experimental conditions in the results section as well.

Discussion

Line 366 – I believe it is overstating your conclusions to say glycerol and sodium induced hydration does not increase body mass. Greater fluid retention is evidence of an increase. Instead, I think it is more accurate to say the magnitude of the typical daily body mass fluctuations within each person was more variable than the observed between conditions differences in body mass change across different participants. But I do not believe it is accurate to say the glycerol and sodium does not increase body mass. If you examine the relative changes in body mass rather than the absolute number, I believe your results would align with the fluid retention values.

Line 388-Good point about the time available before the race.

Line 445 – This appears to be the incorrect reference number.

**Do you want your identity to be public for this peer review?** For information about this choice, including consent withdrawal, please see our Privacy Policy

Reviewer #1: No

Reviewer #2: No

---

## [Author Response · Author response to Decision Letter 1]

5 Dec 2025

**Below is a copied version of our submitted 'Response to Reviewers' document, please see this document for this response in an easier to read format**

Academic Editor’s Comments to Author:

Many thanks for these resources. We have now adjusted the formatting of our manuscript to the PLOS ONE formatting requirements. We have made the following changes:

• We have adjusted all level 1 headings to bold type, 18pt font, and all level 2 headings to bold type, 16pt font.

• We have adjusted headings to be written in sentence case[BP1.1][BP1.2][BP1.3]

• We have adjusted all figure and table titles to use bold type font.

• We have adjusted gastrointestinal symptom tables to include correctly formatted table footnotes

Thank you for these clarifications. This data is now available through Deakin University’s research repository at the following DOI: https://doi.org/10.26187/deakin.29434217.v1

Captions for all supporting information (figures, tables and supplementary tables) have now been provided within lines 694-727 of the revised manuscript.

Thank you for reminding us of this requirement. We have checked all reviewer comments and can confirm that there were no specific works suggested.

Thankyou for this reminder. These checks have now been completed.

Reviewers' Comments to Author:

Reviewer #1

General Comments:

This manuscript addresses an important and timely question regarding hyperhydration strategies in sport and occupational contexts, specifically the combined ingestion of glycerol and sodium bicarbonate. To my knowledge, this is the first study to directly examine the effects of this combination on fluid retention and blood buffering capacity, and the findings offer valuable practical insights for athletes competing in hot environments and situations where pre-event preparation time is limited. The study is well-designed, with a randomized crossover approach, clear methodology, and careful monitoring of hydration, buffering markers, and gastrointestinal symptoms. The results are robust, showing that both glycerol and sodium bicarbonate enhance fluid retention compared to control, and that the combination accelerates retention and enhances buffering capacity without worsening GI symptoms. These findings make a meaningful contribution to the literature on hydration strategies and performance support. Overall, I find the study scientifically sound, ethically conducted, and appropriately reported. With minor adjustments to data availability and interpretation, this manuscript will make a valuable addition to the literature and I recommend it for publication.

Thank you for the time and effort you have spent reviewing our manuscript, it is very much appreciated.

We have provided a response to each of your specific comments and have indicated with text highlighted yellow where we have made changes within the revised manuscript in response.

We appreciate your contributions to our manuscript and feel that your suggestions have improved the revised document.

Specific Comments:

1) Your study combines glycerol with sodium bicarbonate (NaHCO₃) for hyperhydration, which you note has not been directly investigated before. Could you more explicitly highlight what new insights this combination provides compared to the well-studied glycerol + NaCl strategy?

Thank you for this suggestion. We have now more explicitly highlighted the new insights that our investigation of the combination of glycerol with sodium bicarbonate provides compared to the well-studied glycerol + NaCl strategy.

The updated text within the revised manuscript reads:

‘Currently the only combined Na+ and glycerol hyperhydration strategies to be fully investigated have utilised NaCl (10). However, NaHCO3 has recently demonstrated as a source of Na+ to induce hyperhydration (18). Consequently, combining NaHCO3 and glycerol may induce additive fluid retention, similar to that previously reported in combined NaCl and glycerol hyperhydration strategies, while also concurrently improving buffering capacity.’

This change appears on lines 93-96 of the revised manuscript.

2) How does your finding that G+SB accelerates fluid retention (but does not enhance peak retention compared to SB or G alone) advance the field in practical terms? For example, could this be positioned as a novel solution for athletes with limited pre-competition preparation time?

Thank you for raising this point. We have now made a change to the revised manuscript to more clearly explain the practical applications of our findings. The updated text reads:

‘A potential practical application of our findings may be most relevant to the time period before competition, which often limited on the day of major events for endurance events. For instance, early morning starts are common for open water swimming, triathlon, marathon/ultra marathon and race-walking events, and G+SB may therefore be a more feasible hyperhydration strategy if selected over SB or G for athletes competing in these events.’

This change appears on lines 494-500 of the revised manuscript.

3) Could you clarify whether the novelty of your study lies primarily in the physiological insight (timing and buffering interactions) or in the applied context (sport, occupational use), and how this differs from prior work?

Thank you for raising this question. The novelty of our study lies primarily in physiological insight, given that there is no exercise test included in the current study. This study is the first to investigate combined glycerol and NaHCO3 hyperhydration and the effects on hydration status, buffering capacity and gastrointestinal symptoms. To add further clarity regarding the novelty of our study and how this differs from prior work, we have added the following text to the updated manuscript:

‘This is the first study to our knowledge to investigate the combined hyperhydration potential of glycerol and NaHCO3. The novelty of this study lies primarily in the key physiological insights gained into the timing responses of hydration status, blood buffering capacity, and GI symptoms following combined glycerol and NaHCO3 hyperhydration. Future research will however be able to utilise the timing insights gathered from our study and integrate exercise testing.’

This change appears on lines 359-363 of the revised manuscript.

4) The study reports no changes in plasma volume despite significant differences in fluid retention. Could you elaborate on why plasma volume may not have reflected fluid retention differences, and whether methodological sensitivity [BP2.1]may be a factor?

Thank you for this comment. We believe this could be due to slight differences in Hb results reported by different analysers, even though valid methodologies were used in the current investigation and previous investigations. To explicitly elaborate further as to how these differences may have resulted in plasma volume not reflecting fluid retention differences, the following section at lines 420-430 has been added:

‘Some previous studies have reported (as with the current investigation) changes in some but not all hydration measures (18, 42). One potential for differences between the current investigation and that of Siegler et al (who reported a significant change in PV, but no significant change in BM) could be due to slight differences in Hb results reported by different analysers, even though valid methodologies were used in the current investigation (ABL800 Flex, Radiometer) and Siegler et al’s (i-STAT 1 Wireless, Abbott) (18). Additionally, in other previous studies where an increase in ΔPV was reported (5, 10), a different analyser (H2 Photometer, HemoPoint) than that used in the current study was used to generate Hb results. Therefore, future research needs to more thoroughly investigate and compare the results of different analysis methods that can be used to determine Hb.’

5) You used a glycerol dose (1.0 g·kg⁻¹ BM) that is lower than some prior studies. Could you justify this choice more explicitly in terms of expected efficacy versus GI tolerance, and whether higher doses might have altered the outcomes?

Thank you for this suggestion. Isolated sodium bicarbonate and glycerol ingestion has been reported to increase incidence and severity of GI symptoms, therefore a lower dosage of glycerol of 1.0 g·kg⁻¹ BM (a dose which has been demonstrated to elicit improvements in fluid retention), was used to minimise the potential additive effect combined ingestion could have on GI symptoms. To explicitly highlight the efficacy of the glycerol dosage used, the following has been added to the revised manuscript:

‘As both supplements can increase the incidence and severity of GI symptoms following the ingestion of both supplements in isolation, a lower glycerol dosage of 1.0 g.kg-1 BM than most frequently reported in previous studies was utilised during combined supplementation (9, 24, 25). A dose of 1.0 g.kg-1 BM glycerol has been demonstrated to significantly increase fluid retention compared to a fluid bolus ingested alone (25).’

These changes appear in lines 196-201 of the revised manuscript.

To explicitly outline how this may have altered the outcome of results, the following has been added to lines 406-410:

‘The combination of the current NaHCO3 dosage and a higher glycerol dosage (~1.2 g.kg-1 BM) should also be investigated in future research, as this specific combination with the glycerol provided at a higher dose may induce increased fluid retention in glycerol treatments, although the combination needs to be specifically tested as it may also elicit an increased incidence and severity of GI symptoms.’

6) Since no additive benefit was observed in peak fluid retention for G+SB compared to single agents, do you think the practical advantage of G+SB is mainly about timing of onset? If so, would you recommend it over SB alone in competitive practice?

Thank you for this suggestion. This is a valuable point which we agree with and have added more clarity surrounding our suggested use of the SB strategy in our practical application section. We have specified that we recommend SB as a suggested strategy alongside G+SB, when concurrent hydration and buffering is required. The revised text is provided below:

‘The improvements in buffering capacity observed in this study suggest that G+SB and SB may be beneficial in some specific competitive events where adequate hydration status and buffering capacity may be beneficial (e.g. in the context of sprint finishes in road cycling events which also require prolonged effort (64)), or occupational contexts (e.g. shipboard firefighting performed by Navy personnel (65)).

This change appears on line 500-504 of the revised manuscript.

7) You note that GI symptoms were minimal across conditions. Could you provide more context on whether this is consistent with prior SB-only studies, where GI issues are often reported? What might explain the reduced severity in your cohort?

We thank the reviewer for this comment. A comparison of the GI symptoms results from the current investigation with previous SB-only studies has now been added to the revised manuscript.

‘The minimal GI symptoms reported in the current study may be at least partially explained by participants coingesting NaHCO3 with a carbohydrate meal, a strategy reported to reduce the incidence of GI symptoms in comparison to when NaHCO3 is ingested without a carbohydrate meal (37). Early studies which reported GI symptoms would often ingest NaHCO3 in solution and without a coingested carbohydrate meal (59). Over the past two decades however, as the practice of ingesting NaHCO3 via capsules and with a carbohydrate meal has been more frequently utilised, NaHCO3 studies have more commonly reported low incidence and severity GI symptoms (60).’

This change appears on lines 462-470 of the revised manuscript.

Reviewer #2:

General Comments:

In this study, investigators sought to assess the potential additive effects of sodium bicarbonate and glycerol on fluid retention and blood buffering parameters in physically active adults. Overall, fluid retention was increased with glycerol and sodium bicarbonate from 120-180 minutes post ingestion compared to control, promoting a slightly earlier time course for fluid retention compared to sodium bicarbonate and glycerol alone. This reviewer commends the authors for a well thought out study and offer the following recommendations to enhance the manuscript.

Many thanks for accepting the invitation to review our manuscript.

We appreciate the time you have spent reviewing our manuscript, and your expert comments and suggestions.

All changes to the manuscript are indicated with yellow highlighted text.

Specific Comments:

Introduction:

Line 72 add “and” before “increase plasma volume”.

Thank you for identifying this, this change has now been made.

Remove the double citation block on line 76. It is unclear if it should be (7, 17) or (11-16).

Many thanks for bringing this to our attention, this change has now been made.

Remove double citation block in line 80.

We again appreciate you for bringing this to our attention, this change has now been made.

You can separate the paragraphs for each hyperhydration agent.

Thank you for this suggestion, this paragraph has now been broken down three separate paragraphs that focus on 1) water induced hyperhydration, 2) glycerol and sodium as hyperhydration agents, and 3) the introduction to combined ingestion and sodium bicarbonate as a source of sodium.

These updated paragraphs appear on lines 64-107 of the revised manuscript.

The introduction could benefit from a brief discussion of the potential drawbacks of NaHCO3 on gastrointestinal distress, thereby supporting your decision to include this measurement.

---

## [Decision Letter · Decision Letter 1]

5 Jan 2026

Combined glycerol and sodium bicarbonate elicits improvements in fluid retention and blood buffering capacity

PONE-D-25-35201R1

Dear Dr. Patrick,

We’re pleased to inform you that your manuscript has been judged scientifically suitable for publication and will be formally accepted for publication once it meets all outstanding technical requirements.

Kind regards,

William M. Adams

Academic Editor

PLOS One

Additional Editor Comments (optional):

Reviewers' comments:

Reviewer's Responses to Questions

**Comments to the Author**

Reviewer #1: All comments have been addressed

Reviewer #2: All comments have been addressed

2. Is the manuscript technically sound, and do the data support the conclusions?

Reviewer #1: Yes

Reviewer #2: (No Response)

3. Has the statistical analysis been performed appropriately and rigorously?

Reviewer #1: Yes

Reviewer #2: (No Response)

4. Have the authors made all data underlying the findings in their manuscript fully available?

Reviewer #1: Yes

Reviewer #2: (No Response)

5. Is the manuscript presented in an intelligible fashion and written in standard English?

Reviewer #1: Yes

Reviewer #2: (No Response)

Reviewer #1: The authors have answered all of my comments and I wish them good luck with the publishing of the manuscript!

Reviewer #2: (No Response)

**Do you want your identity to be public for this peer review?** For information about this choice, including consent withdrawal, please see our Privacy Policy

Reviewer #1: **Yes:** M AL RIZQI DHARMA FAUZI

Reviewer #2: No

---

## [Editor Report · Acceptance letter]

PONE-D-25-35201R1

PLOS One

Dear Dr. Patrick,

I'm pleased to inform you that your manuscript has been deemed suitable for publication in PLOS One. Congratulations! Your manuscript is now being handed over to our production team.

Kind regards,

on behalf of

Dr. William M. Adams

Academic Editor

PLOS One